# Influence of Abutment Geometry on Zirconia Crown Retention: An In Vitro Study

**DOI:** 10.3390/ma18112469

**Published:** 2025-05-24

**Authors:** Bayandelger Davaatseren, Jae-Sung Kwon, Sangho Eom, Jae Hoon Lee

**Affiliations:** 1Department of Prosthodontics, Yonsei University College of Dentistry, 50-1 Yonsei-Ro, Seodaemun-Gu, Seoul 120-752, Republic of Korea; d.bayandelger@gmail.com; 2Department and Research Institute of Dental Biomaterials and Bioengineering, Yonsei University College of Dentistry, Seoul 120-752, Republic of Korea; jkwon@yuhs.ac; 3HERIBio Co., Ltd., Seoul 158-867, Republic of Korea; heribio@icloud.com

**Keywords:** Ti-base abutments, abutment geometry, CAD/CAM, pull-out test, retention

## Abstract

**Background/Objectives**: This in vitro study investigated the retention of three different geometrical designs of short titanium base (Ti-base) abutments used in implant-supported zirconia crowns. The advent of digital technology has facilitated the integration of Ti-base abutments into implant dentistry by improving time efficiency, precision, and patient comfort. **Methods**: Three types of short Ti-base abutments were evaluated: Geo SRN multibase^®^ (Group A), Herilink^®^ (Group B), and TS Link^®^ (Group C), each with a height of 4 mm and gingival height of 1 mm (n = 20 per group). Zirconia crowns (LUXEN^®^ Smile S2, DentalMax, Republic of Korea) were modified for the testing setup and fabricated using CAD/CAM technology, then bonded to the abutments with RelyX^®^ Luting 2 resin-modified glass ionomer cement. Pull-out tests were conducted at a crosshead speed of 1 mm/min to assess retention. **Results**: One-way ANOVA and post hoc Tukey tests revealed significant differences in retention values among the different abutment shapes (*p* < 0.05). The mean retention forces were 194.65 N for Group A, 241.33 N for Group C, and 360.20 N for Group B. **Conclusions**: The geometrical design of Ti-base short abutments significantly affects the retention of CAD/CAM zirconia crowns, with hexagonal shapes (Group B) demonstrating superior retention. Clinically, selecting an abutment design with enhanced mechanical retention may improve the long-term success of implant-supported restorations.

## 1. Introduction

Implant-supported restorations are a dependable treatment option for dentists when dealing with a single missing posterior tooth [1]. Advancements in implant design, surface treatments, prosthetic materials, and surgical guidance have significantly improved the survival rates of dental implants [2]. A systematic review has reported survival rates of over 97% across all placement and loading protocols [3]. Additionally, reviews suggest a 97.6% survival rate for single-implant restorations without complications after three years [4].

Various materials are available for the fabrication of implant-supported single crowns [5]. In implant treatment, titanium abutments paired with porcelain-fused-to-metal (PFM) crowns are considered the gold standard, boasting a five-year survival rate of approximately 98.3% [6]. To enhance aesthetics, zirconia-based restorations have been introduced [7]. Monolithic zirconia, manufactured using computer-aided design and computer-aided manufacturing (CAD/CAM), offers high biocompatibility and reduces plaque accumulation [8]. In the posterior region, where high occlusal forces are present, materials with strong fracture resistance are essential. Zirconia has consistently proven effective for abutments and implant restorations in these areas and offers a similarly effective alternative to PFM crowns for restoring single implants in the posterior region [9].

Implant prostheses are typically fixed to standard or custom abutments using either screws or cement [10,11]. Cement-retained restorations offer a superior aesthetics by eliminating the need for a screw hole, which also removes the need for composite resin repairs and allows for more effective ceramic layering. These restorations are also believed to withstand occlusal forces better due to the enhanced ceramic layering. Cement-retained restorations are often preferred when implants are placed at an angle that deviates from the ideal prosthetic axis [11]. However, a major drawback of cement-retained restorations is the difficulty in removing excess cement from the gingival sulcus, which can lead to peri-implantitis and make the restoration irretrievable [12,13,14]. On the other hand, screw-retained restorations offer the advantage of retrievability and avoid biological complications related to cementation. Screw-retained crowns are associated with fewer pathogenic bacteria when compared with cemented crowns, which have more inflammatory cells and a higher presence of periodontal pathogens [15].

Currently, digital technology is playing an increasingly important role in oral implantology. Computer-aided technology has transformed dentistry by providing efficient and precise methods for creating various restorations, including implants. The rise of digital workflows in dentistry has led to a growing interest in Ti-base abutments [16]. These advancements have enabled a fully digital workflow in oral rehabilitation, where intraoral scanning and CAD/CAM technology allow for faster, more precise processes, improving time efficiency by around 50% and ensuring highly accurate prosthetic reconstructions [17,18]. The digital impression scanning significantly reduces the need for traditional alginate or silicone rubber impressions, minimizing patient discomfort such as nausea and, in some cases, gagging. However, conventional impressions may still be necessary in certain clinical situations. Ti-base abutments are compatible with CAD/CAM, allowing for the quick creation of well-fitting prostheses. 

Ti-base abutments are pre-made titanium components with a hybrid design that allows both cemented and screw-retained fixation within a single prosthesis [19,20]. These abutments connect either to monolithic customized crown or to a high-strength customized ceramic abutment with a cement-retained crown (a three-piece screwed restoration) [21]. This hybrid retention mechanism makes it easier to remove excess cement and ensures improved light curing of the restoration margins before screwing in the final restoration. CAD/CAM-generated restorations, such as zirconia crowns or abutments, can be cemented onto these abutments, enhancing both their versatility and reliability. One of the key advantages of Ti-base abutments is their retrievability. Like UCLA abutments and other screw-retained systems, Ti-base abutments allow the abutment–crown assembly to be cemented outside of the mouth, ensuring excess cement is removed to prevent peri-implantitis before final screw fixation [13]. Furthermore, CAD/CAM systems now include comprehensive libraries for the rapid fabrication of prostheses using Ti-base abutments [20].

The selection and cementation protocols for Ti-base abutments vary, and choosing the right abutment is crucial to the success of implant treatment. Many companies now offer Ti-base abutments specifically designed for digital dentistry. Manufacturers have improved the geometry and design of these abutments to enhance retention, making even short abutments a viable option for restoring edentulous spaces. The retention of the final prosthesis is influenced by several factors, including the height of the Ti-base, the surface texture, the type of cement used, the fit of the superstructure, and any surface treatments applied [20]. Achieving adequate retention with short abutments remains a challenge, and companies provide various abutment designs for digitally created prostheses. However, there is currently limited research on the geometric differences among the various types of Ti-base abutments.

This study aims to compare the retention of three different shapes of short Ti-base abutments for implant crown restoration. The primary objective of this study is to evaluate whether the geometric shape of Ti-base abutments influences the retention of CAD/CAM zirconia crowns. The null hypothesis tested was that different abutment geometries would not significantly affect crown retention. However, we hypothesized that abutment shape has a significant effect. This research intends to provide valuable insights for clinicians in selecting the most appropriate implant abutment, ultimately improving the success of implant-supported restorations in cases with limited vertical space.

## 2. Materials and Methods

### 2.1. Sample Preparation

In this in vitro study, 60 test specimens were evaluated using three distinct types of short Ti-base abutments (n = 20/group). Figure 1 illustrates the specimens used in this experiment. The experimental method used in this study was based on previously validated protocols for assessing crown retention forces [22,23], with modification to suit the geometrical variations of the abutments evaluated. These abutments were divided into three groups based on their shape.

Ti-base abutments were screwed to an implant analog (GSTLA400, TS Fixture Lab analog, Osstem, Seoul, Republic of Korea) and tightened to the manufacturer recommended torque of 30 Ncm using a torque wrench [22,24]. All screw channels were closed with Teflon tape.

### 2.2. Scanning and Crown Fabrication

Zirconia copings were modified for the test setup. Each group of abutments was scanned using a desktop scanner (E1 3shape TRIOS 3D model scanner, København, Denmark) and were designed using software (Dental System Premium^®^, 3Shape, København, Denmark, https://www.3shape.com/en/software/dental-system, accessed on 30 April 2025) [18,19]. Figure 2 and Figure 3A show the three-dimensional design. The superstructure was planned as a spherical-shaped zirconia coping with height of 7.5 mm, a width of 9 mm, and a diameter of 6 mm. The cement gap was set at 40 μm [25,26], which is similar to the values used in clinical settings. Zirconia copings were milled from prefabricated discs (LUXEN Smile S2 zirconia block, DentalMax, Seoul, Republic of Korea) and subsequently sintered at 1350 °C for 7 h. The marginal fit was evaluated under 4× magnification. The sample size was determined based on previous similar studies and power analysis (α = 0.05, power = 0.8), ensuring sufficient statistical strength [16,22,24].

### 2.3. Cementation

CAD/CAM-generated zirconia copings, as shown in Figure 3B, were bonded to geometrically different abutments using RelyX^®^ resin-modified glass ionomer cement (3M ESPE^®^, St. Paul, MN, USA). All specimens were cemented by the same operator, following the manufacturer’s instructions. The specimens were then stored at room temperature for 30 min until the complete setting reaction had occurred. After the setting period, all excess cement was removed with an explorer, and each surface was polymerized for 60 s using an LED curing light (Elipar DeepCure-S, 3M ESPE, St. Paul, MN, USA) according to the manufacturer’s instructions. The specimens were stored at room temperature for 30 min after cementation to ensure complete setting before conducting the pull-out tests.

### 2.4. Pull out Test

The pull-out test method was chosen for this study as it is a well-established and validated approach for assessing the retentive force between crowns and abutments, particularly in relation to geometric design [22,23]. The specimens were assembled, as shown in Figure 4, in a universal testing machine (5942 Model, Norwood, MA, USA) and subjected to a pull-out test (retention) at a crosshead speed of 1 mm/min. The force required to remove the copings was recorded in newtons and tabulated for statistical analysis. The time to crown separation was not fixed and depended on the retention force of each specimen. Each coping was dislodged once the applied tensile force exceeded the bond strength, which occurred within a few seconds under the constant testing speed (Figure 5).

### 2.5. Statistical Analysis

Statistical analysis was performed using IBM SPSS Statistic software, version 29.0 (IBM Corp., Armonk, NY, USA). A one-way analysis of variance (ANOVA) was conducted to compare the mean retention forces between the three groups of Ti-base abutments. When significant differences were found (*p* < 0.05), post hoc multiple comparisons were performed using Tukey’s honestly significant difference (HSD) test. A significance level of α = 0.05 was set for all tests. Data were presented with mean values, and 95% confidence intervals (CI) were calculated to assess the precision of the estimates.

## 3. Results

The mean tensile force required to separate the copings from the abutments is shown in Figure 6. The mean retention forces were 194.65 N for Group A, 360.20 N for Group B, and 241.33 N for Group C.

A one-way ANOVA revealed a statistically significant differences in retention forces among the three groups (*p* < 0.001) (Table 1). Post hoc Tukey’s HSD tests showed that Group B (hexagonal-shaped abutment) demonstrated a statistically higher retention force when compared with both Groups A and C (*p* < 0.001) (Table 2). 

Adding grooves to the cylindrical Ti-base design resulted in an increase in mean retention force from 194.65 N in Group A to 241.33 N in Group C, although the difference was not statistically significant. These results suggest that abutment shape, particularly the hexagonal design, has a substantial effect on the retention strength of CAD/CAM zirconia copings.

## 4. Discussion

In the present study, retention forces varied among the three tested abutments. Therefore, the first null hypothesis, which suggested that there would be no significant differences in retention among short Ti-base abutments of different shapes, was rejected. The hexagonal shaped Ti-base abutment demonstrated a higher pull-out force when compared with the circumferential shaped Ti-base abutment. The long-term success of implant-supported restorations depends on several factors, including abutment design, taper angle, height, texture, cement type, and surface pretreatment [16,19]. Selection of the appropriate abutment is crucial in preventing complications with implant-supported restorations. One main advantage of Ti-base abutments, as previously mentioned, is the ability to perform the bonding procedure before crown placement.

In terms of abutment height, 4 mm abutments were used in this study. Using short titanium base abutments in the posterior region is often recommended due to the limited interocclusal space commonly found in edentulous patients [22]. The reduced space in this area can make it challenging for dentists to achieve optimal retention for prostheses. Short Ti-base abutments help address this issue by providing a more practical solution for securing restorations in cases where vertical height is limited, ensuring better stability and retention without compromising the prostheses design. The hexagonal Ti-base abutment, with retentive elements and 4 mm height, increased the retention of the zirconia restoration. If the available restorative space allows for the use of Ti-base abutments taller than 4 mm, a taller abutment is recommended to improve retention [16]. However, in cases where the restorative space is limited, a 4 mm abutment may be necessary. Knowing the minimum height requirement for Ti-base abutments can be valuable during pre-surgical planning to ensure there is enough vertical space to accommodate this minimum height. Previous studies have shown that abutment height can influence the retention of implant-supported restorations, though results have been inconsistent [24]. One study found that crown material and Ti-base height had a significant effect, but their interaction was not significant [27]. Increasing the height of Ti-base abutment significantly improves the retention of zirconia restoration [22]. However, some researchers have indicated that abutment height has less impact on retention than the abutment geometry [24].

Few studies have considered the geometric aspect of an implant abutment. The geometric shape of an abutment has a crucial role in retention besides bond forces and micro retentive forces [28]. Adding circumferential grooves to implant abutments increased the retention of cement-retained restorations [23]. Manufacturer is important; for both Zr and Ti-base abutments, parts from different manufacturers, design, and manufacturing differences influenced performance and appeared extremely similar on clinical examination [19]. Finally, design defect problems were suggested for all systems. Once again, the manufacturer matters; differences in design and fabrication that influence performance cannot be discerned clinically [19,29].

In this study, the cement gap size was set to 40 μm. Increasing the cement gap from 30 to 60 μm negatively impacted cement durability [25]. Increasing the cement gap from 10 to 50 μm significantly reduced the pull-off force of implant-supported crowns [26]. Most studies recommend a cement thickness of 30 to 40 μm to ensure complete seating of the restoration. Based on this, a cement gap of 40 μm was selected for this study, with permanent cement.

The height of the abutment and convergence angle play critical roles in the retrievability of prostheses. Studies have shown that, as the convergence angle increases and abutment height remains constant, removal torque values decrease significantly [30]. These findings align with previous research highlighting the impact of abutment height and surface area on prosthesis retention. Clinically, the results of this study suggest that hexagonal Ti-base abutments may be preferred in cases where enhanced retention is required, such as in posterior regions with limited interocclusal space or when retrievability is critical. This may help improve the long-term stability of restorations under functional loading. It is also crucial to consider the surface area and total occlusal convergence (TOC) of the abutments, as these factors greatly influence the retention of cement-retained prostheses. Research indicates that maximum retention in full-veneer crowns is achieved with parallel axial walls, while retention decreases substantially as the TOC angle increases [31]. Additionally, another study found that cylindrical preparations with a 20-degree TOC provide greater retention when compared with those with a 30-degree TOC [32]. 

The limitations of this study include its in vitro design, which allows for the evaluation of a specific variable—geometric design. Although uniaxial pull-out retention tests do not perfectly replicate intraoral conditions, they are an efficient alternative for determining the retentive force between an abutment and a zirconia restoration. Based on this study, it can be inferred that all tested groups could withstand average physiological occlusal forces typically exerted in the posterior molar region. However, intraoral occlusal forces are dynamic rather than static. In vitro static pull-out tests may not fully replicate the dynamic oral environment, where masticatory forces, saliva, and thermal cycling can influence long-term retention outcomes. Additionally, other factors, such as surface treatment and abutment material composition, may impact retention and should be considered in future studies. Further, in vitro and clinical studies are necessary to determine a universal abutment selection protocol for optimal restoration retention.

## 5. Conclusions

The purpose of this study was to assess how different geometric variations in the dental implant Ti-base abutments influence the retention of cemented implant-supported prostheses. The hexagonal design of the Ti-base abutment could be increased the retention of implant-supported prosthesis. Additionally, the geometrical design of the abutments affected the retention of zirconia crowns, with some shapes showing better retention characteristics than others. When interocclusal space is limited, a hexagonal Ti-base is recommended due to its enhanced retention. Further in vitro and clinical studies are recommended to validate these findings under dynamic loading conditions and to develop guidelines for optimal abutment selection.

## Figures and Tables

**Figure 1 materials-18-02469-f001:**
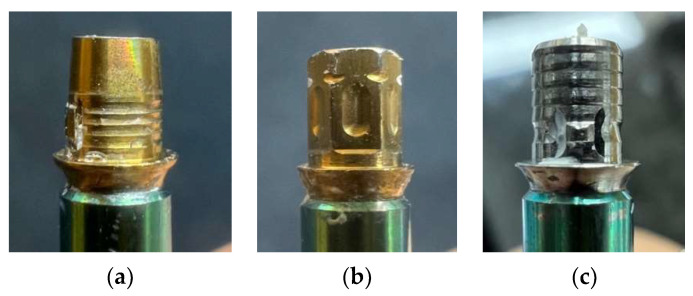
Specimen illustration used in this experiment. (**a**) Group A—diameter 4.5 mm, abutment height 4 mm, gingival height 1.2 mm, total height 5.2 mm. This group features a cylindrical shape that slightly tapers in the second half from the midpoint. The lower part has grooves around the cylindrical surface, and there is a rectangular ledge protruding from the abutment surface (Geo SRN multibase^®^ abutment, Geo Medi, Seoul, Republic of Korea). (**b**) Group B—hexagonal shape with across flats (WAFs) of 4 mm, abutment height 4 mm, gingival height 1.3 mm, total height 5.3 mm, and a 0° convergence angle. The hexagonal cylinder features rectangular dimples on each face and grooves at the top of the vertices (Herilink^®^ abutment, Heri Implant, Seoul, Republic of Korea). (**c**) Group C—diameter 4.5 mm, abutment height 4 mm, gingival height 1 mm, total height 5.0 mm, with a 0° convergence angle. This group features a cylindrical shape with numerous grooves around the abutment surface and there is a rectangular ledge located just above the flat form (TS Link^®^ abutment, Osstem, Seoul, Republic of Korea).

**Figure 2 materials-18-02469-f002:**
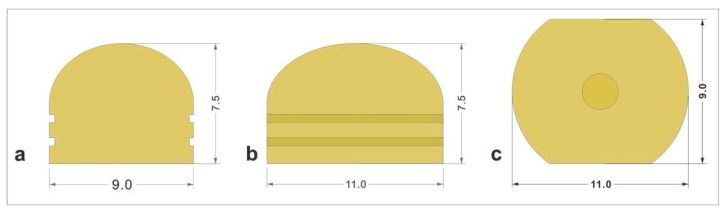
Modified zirconia coping diagram used in this experiment. (**a**) Frontal view; (**b**) lateral view; (**c**) lower view.

**Figure 3 materials-18-02469-f003:**
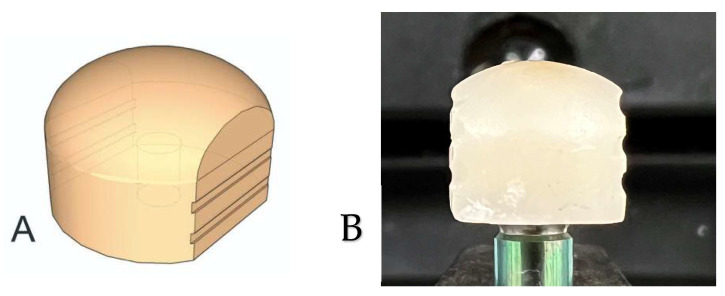
(**A**) Three-dimensional design of the modified zirconia coping and (**B**) the fabricated zirconia coping.

**Figure 4 materials-18-02469-f004:**
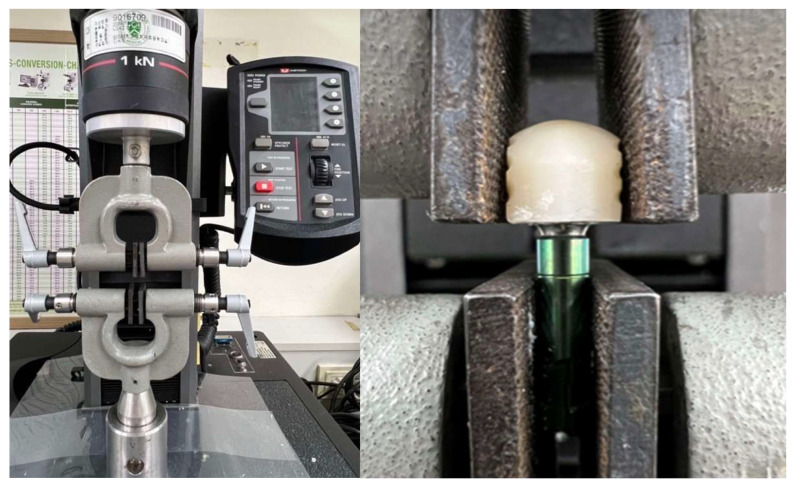
The universal testing machine (5942 model, Norwood, MA, USA) used for measuring pull-out force at a crosshead speed of 1 mm/min in this experiment.

**Figure 5 materials-18-02469-f005:**
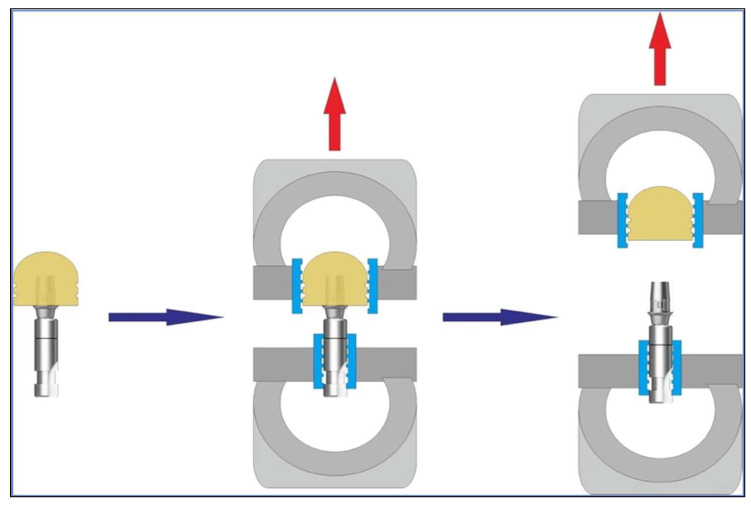
Experimental design: Ti-base abutment and zirconia crown design attached for the pull-out test to measure retentive strength.

**Figure 6 materials-18-02469-f006:**
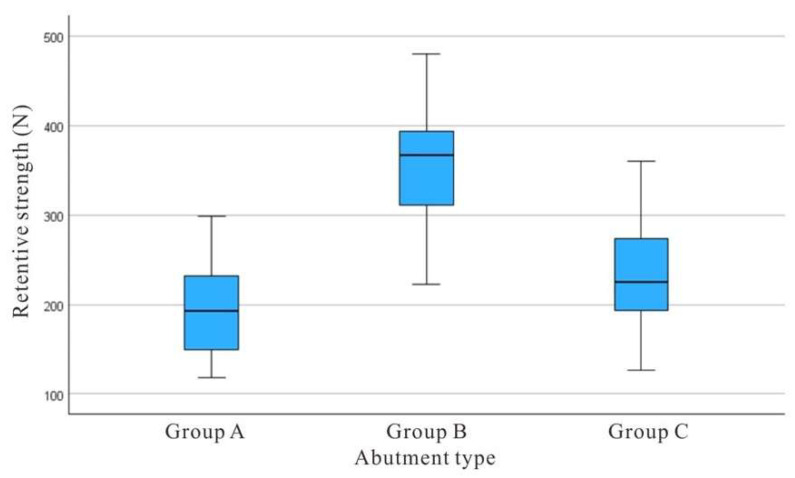
Box-plot diagram showing the retention strength of different abutment groups (in Newtons).

**Table 1 materials-18-02469-t001:** One-way ANOVA results showing differences in retention forces among three Ti-base abutment groups.

Oneway ANOVA					
Data	Sum of Squares	df	Mean Square	F	Sig.
Between groups	291,448.65	2	145,624.32	28.504	<0.001

**Table 2 materials-18-02469-t002:** Post hoc Tukey’s HSD test results for pairwise comparisons of mean retention forces between Ti-base abutment groups.

Group	Mean Difference	Standard Error	Sig.	95% Confidence Interval	
				Lower Bound	Upper Bound
Group (B–A)	165.55	22.61	<0.001	111.14	219.96
Group (B–C)	118.86	22.61	<0.001	64.45	173.27
Group (A–B)	46.68	22.61	0.106	7.72	101.09

The mean difference is significant at the 0.05 level.

## Data Availability

The original contributions presented in this study are included in the article. Further inquiries can be directed to the corresponding author.

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
