# Peer review of "Influence of Abutment Geometry on Zirconia Crown Retention: An In Vitro Study"

_materials, 2025, doi:10.3390/ma18112469_

Round 1

Reviewer 1 Report

Comments and Suggestions for Authors

see attached pdf

Reviewer 2 Report

Comments and Suggestions for Authors

This article addresses the topic of the influence of the macroscopic morphology of the abutments on the retention of zirconia prostheses.

Abstract

This section is correct and reports the summary of the main aspects of the paper

Introduction

Various materials have been used for the manufacture of prosthetic abutments and crowns on implants. Materials such as titanium, zirconium, and ceramics have played an important role.

The design of the abutments is also essential to facilitate the fixation of the prosthesis to the implant. Computer-aided technology providing efficient and precise methods for creating various restorations, including implants.

This study aims to compare the retention of three different shapes of short Ti-base abutments for implant crown restoration.

This section is correct and reports updated scientific information on the topic.

Material and methods

The authors present the methodology for conducting the study; the information is precise and concise, and the images are of good quality.

The authors should report the total height of the abutments, since it is also important in the retention of the prostheses.

Authors should include the symbol ® after each trade name of both the experiments and the different devices used in the study, as is standard in scientific papers.

The methodology is adequate to assess the study's objective. However, the authors could improve the quality of this section by providing references to scientific papers that support the methodology's various procedures.

The information on the statistical analysis of the results needs to be completed. It is too simplistic and does not provide the more detailed information needed for future readers to understand.

Results

The results are expressed in an overly concise manner, as is the statistical significance. Perhaps the authors could have presented them in a table for better understanding and comparison.

Discussion

The authors discuss the study's results, comparing them with those obtained in other international studies. The authors should improve the discussion by providing more and more up-to-date references (within the last 5 years), as there are some paragraphs with a wealth of information but few or no references.

In addition, the authors should present further information and discussion on the clinical application of the study.

Conclusions

This section is correct

References

This section should be reviewed. Authors present references with a single author and the term "et al." This format is incorrect and does not follow the standards for scientific journals.

Reviewer 3 Report

Comments and Suggestions for Authors

Interesting idea for research, but it seems to me that this is only the beginning of a certain research series, because different cemeteries can connect abutment materials of different shapes in different ways, so time has a great influence on the strength of the connection.

The article is well written, it only requires a few corrections

Abstract: It would be good to add what zirconium oxide the crowns were made of (material name)

Introduction

Line 72

The digital impression scanning also eliminates the need for traditional alginate  or silicone rubber impressions, avoiding patient discomfort such as nausea and gagging- this is true for some cases, because we all know that it is not possible to eliminate impression materials in all cases

Before the tests, should I put forward some thesis, e.g. that the shape of the abutment influences crown retention?

M&M

Is the research method you used your idea or has it already been described by someone? I mean is it validated in any way?

Line 156

After the setting period, all  excess cement was removed with an explorer, and each surface was polymerized for 60 seconds.- What instrument was it hardened with? polymerization lamp (producer type) why 60 sec?

How long did it take for the crown to be separated from the abutment?

The same tests could be repeated using an accelerated aging method, e.g. thermocycling.

Discussion I like it

Good luck with your further research!

Round 2

Reviewer 2 Report

Comments and Suggestions for Authors

The review is correct. The authors have modified the manuscript according to the reviewer's recommendations to improve its scientific quality.

Comments on the Quality of English Language

I am not expertise